# The Effect of Cerclage Banding Distally to a Clamshell Fracture Pattern in Total Hip Arthroplasty—A Biomechanical Study

**DOI:** 10.3390/bioengineering10121397

**Published:** 2023-12-06

**Authors:** Philipp Kastner, Ivan Zderic, Boyko Gueorguiev, Torsten Pastor, Matthias Luger, Tobias Gotterbarm, Clemens Schopper

**Affiliations:** 1AO Research Institute Davos, Clavadelerstrasse 8, 7270 Davos, Switzerland; ivan.zderic@aofoundation.org (I.Z.); boyko.gueorguiev@aofoundation.org (B.G.); torsten.pastor@luks.ch (T.P.); 2Department for Orthopaedics and Traumatology, Kepler University Hospital GmbH, Johannes Kepler University Linz, Krankenhausstrasse 9, 4020 Linz, Austria; matthias.luger@kepleruniklinikum.at (M.L.); tobias.gotterbarm@kepleruniklinikum.at (T.G.); clemens.schopper@kepleruniklinikum.at (C.S.); 3Department of Orthopaedic and Trauma Surgery, Lucerne Cantonal Hospital, 6000 Lucerne, Switzerland

**Keywords:** total hip arthroplasty, cerclage wiring, periprosthetic femoral fractures, clamshell fracture, biomechanics, Vancouver classification

## Abstract

Objectives: As currently there is no existing biomechanical work on the topic of interest, the aim of the current study was to investigate the effect of cerclage banding distally to an intraoperatively occurring proximal periprosthetic femoral clamshell fracture versus a non-fractured femur after total hip arthroplasty. Methods: A diaphyseal anchoring stem was implanted in twenty paired human cadaveric femora, assigned to a treatment and a control group. In the treatment group, each specimen was fitted with a cerclage band placed 3 mm distally to a clamshell fracture, created with an extent of 40% of the anchoring length of the stem. The resulting fragment was not treated further. The contralateral specimens were left with the stems without further fracture creation or treatment. All constructs were tested under progressively increasing cyclic axial loading until failure. Relative bone-implant movements were monitored by motion tracking. Results: Number of cycles and the corresponding load at stem loosening, defined as 1 mm displacement of the stem along the shaft axis, were 31,417 ± 8870 and 3641.7 ± 887 N in the control group, and 26,944 ± 11,706 and 3194.4 ± 1170.6 N in the treatment group, respectively, with no significant differences between them, *p* = 0.106. Conclusion: From a biomechanical perspective, cerclage banding distally to an intraoperative clamshell fracture with an extent of 40% of the anchoring length of the stem demonstrated comparable resistance against hip stem loosening versus a non-fractured femur. It may therefore represent a valid treatment option to restore the full axial stability of a diaphyseal anchoring stem. In addition, it may be considered to keep the medial wall fragment unfixed, thus saving operative time and minimizing associated risks.

## 1. Introduction

Periprosthetic fractures (PPFs) are a rapidly increasing problem in total hip arthroplasty (THA) [1,2]. High economic costs, difficult surgical treatment, and poor clinical outcomes associated with high mortality rates render PPFs among the most devastating complications of THAs [3,4,5]. Their incidence ranges within 0.1–1% following cemented THA, and reaches up to 5.4% following uncemented THA [6,7]. Risk factors include the conduction of minimally invasive techniques, a patient age below 50 years and above 80 years, use of uncemented press fit stems, bone loss or disease, as well as technical difficulties [8,9,10,11,12,13]. PPFs most frequently occur intraoperatively during insertion of the stem and often involve the medial wall of the femur [14,15].

In recent years, the importance of the medial wall has drawn increasing attention concomitantly with the significantly increased use of metaphyseal anchoring shafts in cementless stem fixation [16], because it is a crucial contributor to the axial stability of the proximal femur. Similarly, the medial femoral wall has been demonstrated as extremely important for the clinical outcome following fixation of proximal femoral fractures [17,18,19,20]. The part of the femur that includes the lesser trochanter is commonly affected in case of a PPF.

In 2014, the term “clamshell fracture” was introduced, defining a fracture of the lesser trochanter with inclusion of a segment of the proximal medial femoral cortex [21]. In 2018, the Vancouver classification was renewed by Huang et al., and this fracture pattern was added to it [22]. Until then, the outdated Vancouver classification did not actually distinguish between a simple avulsion fracture of the lesser trochanter and a more devastating fracture involving parts of the medial femoral wall [21,22,23,24,25].

It is already known that a clamshell fracture occurs intraoperatively when inserting anatomically shaped press fit stems [26]. Similarly, it is clear that depending on the size of the medial wall fragment, surgical intervention may be required as it belongs to the Vancouver B fractures and consequently should be treated as such. In such a case, a diaphyseal anchoring stem should be preferred over a metaphyseal anchoring one, because the latter provides limited stability due to the fracture pattern [22]. To date, there is no consensus on the further treatment of the medial wall fragment. Despite some recommendations and empirical reported values, no clear guidelines have been set, nor have biomechanical experiments been conducted yet in this regard. Therefore, it remains unclear whether refixation is worth the risk of entailing and whether it contributes to a significant improvement in stability. The anatomical location of the lesser trochanter and the proximity of the medial wall to vessels and nervous structures make the refixation of the medial wall fragment a risky and time-consuming endeavor. Likewise, a significant amount of time is associated with the refixation, which could drastically prolong the operating time and therefore lead to increased blood loss and risk of infection [27]. For intraoperative periprosthetic fractures in the context of THA, an additional “safety” cerclage is often placed around the femoral shaft distally to the fragment to prevent the extension of the fracture and an increase in instability. This cerclage wire can usually be applied without great effort, due to the easily accessible anatomical location and the good intraoperative visibility [28]. CoCr cable cerclages and hose clamps have demonstrated the highest initial stiffness and the lowest decrease in stiffness with increasing load [10]. To date, there are no existing studies investigating the effect of such a cerclage on the axial stability of the stem in case of a periprosthetic femoral clamshell fracture.

Therefore, the aim of this study was to biomechanically investigate the effect of cerclage banding 3 mm distally to an intraoperatively occurring clamshell fracture in a human cadaveric model. It was hypothesized that a cerclage band alone, placed distally to the fracture pattern, would regain full stability compared to a non-fractured femur following THA, and therefore would demonstrate a sufficient stand-alone treatment superseding a refixation of the medial wall fragment.

## 2. Materials and Methods

### 2.1. Specimens and Preparation

Ten pairs of fresh-frozen (−20 °C) human cadaveric femora from three female and seven male donors, aged 77 ± 10.2 years (mean value ± standard deviation; range 59–96 years), were used. No bony lesions, malignancy, or metabolic disease were detectable in the specimens by visual and radiographic examination. High-resolution peripheral quantitative computed tomography (XtremeCT, Scanco Medical AG, Brüttisellen, Switzerland) was used to determine the bone mineral density (BMD) before the start of the study to exclude potential extreme differences between the left and right femora of each pair. The specimens were assigned to two paired study groups—a control and a treatment group—with equal number of left and right anatomical sites. Prior to instrumentation, all specimens were thawed at room temperature, and their soft tissue was removed by an experienced surgeon. In addition, anteroposterior X-rays were taken (C-arm, Arcadis Varic, Siemens Healthineers, Erlangen, Germany), and the required sizes of the hip stems were measured individually for each femur using a planning program (mediCAD, Altdorf, Germany). A new calcium phosphate titanium plasma-coated diaphyseal anchoring press fit stem (ANA.NOVA SL-complete, Ti6Al4V alloy, 127° CCD, double conus design, polished taper; ImplanTec, Mödling, Austria) was used for each bone (Figure 1). The stems were inserted according to the manufacturer’s guidelines, using the press fit technique (Figure 2). The instrumentation of the stems was performed under X-rays control.

In the treatment group, an oscillating saw was used to create the fracture before implantation of the stem. The fracture model was chosen based on a study by Andriamananaivo et al. describing the Remaining Attachment Index (RAI). The RAI sets a 2:3 ratio in favor of the anchoring section as the cut-off for stability to distinguish between Vancouver B1 and B2 fractures [29]. A standardized procedure was conducted as follows. The fracture model was with an extent of 40% of the medial anchorage length of the stem and was therefore dependent on the size of the latter. For each used stem size, the length of the medial anchorage distance was measured with the appropriate rasp, and the desired anchorage loss of 40% was then marked on the rasp. The serrations on the rasp up to the mark were then counted to allow visualization using X-rays. Finally, during instrumentation, the calculated height could be easily determined using a K-wire under fluoroscopy (Figure 3A). Furthermore, the dorso-cranial border of the lesser trochanter (Figure 3B) and the ventral border of the final rasp (Figure 3C) were considered as anatomical landmarks. Thus, in combination with the determined craniocaudal dimension according to the method described above (Figure 3A), all parameters required for setting of the final fracture pattern were defined. The fracture model contained the entire lesser trochanter and included the pre-defined portion of the surrounding medial cortical wall, thus resulting in a proximal medial wall defect around the diaphyseal anchoring press fit stem (Figure 3D). The model was based on a study by Kastner et al. and has been previously used in a biomechanical investigation [30]. After creation of the clamshell pattern, a cerclage band was placed 3 mm distally to the fracture (CCG^®^ band; ImplanTec, Mödling, Austria) (Figure 3D). Fluoroscopy was then used to determine the tip of the stems, and the femora were cut 85 mm distally to them.

Optical marker sets and markers were attached to the femoral shaft and glued on the prosthesis below the ceramic head for optical motion tracking, respectively. Finally, the femora were embedded in 65 mm deep Polymethylmethacrylate (PMMA, SCS-Beracryl D28, Suter Kunststoffe AG, Fraubrunnen, Switzerland) base.

### 2.2. Biomechanical Testing

A Bionix 858 servo-hydraulic test system (MTS Systems, Eden Paririe, MN, USA) equipped with a 25 kN/250 Nm load cell was used for biomechanical testing. The test setup and loading protocol were adopted from previous studies [31,32]. The specimens were mounted in 20° adduction relative to the machine axis [33]. The bearing couple consisted of a 36 mm ceramic head and a steel indenter. To allow for artefactual shear force and moment correction, a cardan joint was interconnected distally between the femur and the machine base (Figure 4).

The loading protocol consisted of a quasi-static and a cyclic loading section. First, non-destructive quasi-static ramped compressive loading was applied between 50 and 200 N at a rate of 15 N/s. Second, destructive cyclic testing with a physiological dual-peak compression profile of each cycle—reflecting the course of hip joint reaction forces during walking as demonstrated by Bergmann et al.—was run at 2 Hz [34]. Whereas the valley load of each cycle was kept constant at 200 N, the peak load, starting at 500 N, was monotonically increased at a rate of 0.1 N/cycle. Consequently, the loading amplitude was steadily increased until test stop, defined by 30 mm axial displacement of the machine transducer relative to the start of the test. The method of progressively increasing cyclic loading to failure has been demonstrated as useful in previous studies [35,36].

### 2.3. Data Acquisition and Evaluation

Machine data in terms of axial displacement and axial load were continuously recorded at 128 Hz from the machine transducer and the load cell, respectively. Based on this, cycles, load, and multiple of body weight at catastrophic failure were evaluated. Catastrophic failure was defined at an axial displacement of 3 mm and indicated by a rapid drop in the load–displacement curve.

The positions of the optical markers attached to the hip stem and the femoral shaft were continuously recorded at 50 Hz with a stereometric measurement system (Aramis, Carl Zeiss GOM Metrology GmbH, Braunschweig, Germany) using two high-resolution optical cameras operating at a resolution of 12 megapixels. Based on the motion tracking data, the relative movements of the stem to the femoral shaft were evaluated in six degrees of freedom. Thereby, the movement of the stem along the femoral shaft axis, plotted over the course of cycles, was considered for assessment of stem loosening. An abrupt change in the trend of this graph was an indicator for the loosening. Because this change consistently occurred at a margin of approximately 1 mm, this value was set as a criterion for stem loosening. This criterion is also consistent with the data from previous studies where 1 mm axial displacement is described as the approximate point at which the stem loosens. The number of cycles until stem loosening was calculated together with the corresponding peak load at that specific time point. Furthermore, based on each donor’s weight, the multiple of body weight at the time point of stem loosening was calculated as follows: (multiple of body weight at stem loosening) = 100*(peak load at stem loosening)/(body weight).

Statistical analysis among the parameters of interest was performed using SPSS (V.27, IBM, Armonk, NY, USA). A priori power analysis revealed that a sample size of at least 10 specimens per group was required for a statistical power of 0.8 at a significance level of 0.05, if the standard deviation of each group did not exceed 110% of the difference in mean values between groups. Mean and standard deviation (SD) values were calculated for each parameter of interest and group separately. Normal distribution of the data was confirmed using the Shapiro–Wilk test. Statistical differences between the two study groups were elucidated with Paired-Samples T-test. The level of significance was set at 0.05 for all statistical tests.

## 3. Results

### 3.1. Cycles and Load at Stem Loosening

Number of cycles and corresponding load at stem loosening were 31,417 ± 8870 and 3641.7 ± 887 N in the control group, and 26,944 ± 11,706 and 3194.4 ± 1170.6 N in the treatment group, respectively, with no significant differences between them (*p* = 0.106) (Figure 5).

### 3.2. Cycles and Load at Catastrophic Failure

Number of cycles and corresponding load at catastrophic failure were 38,714 ± 10,018 and 4371.4 ± 1001.8 N in the control group, and 34,441 ± 10,618 and 3944.1 ± 1061.8 N in the treatment group, respectively, with no significant differences between them (*p* = 0.105) (Figure 6).

### 3.3. Multiple of Body Weight at Stem Loosening

Multiple of body weight at implant loosening was 559.4 ± 166.2% in the control group and 480.9 ± 156.9% in the treatment group (*p* = 0.052).

### 3.4. Multiple of Body Weight at Catastrophic Failure

Multiple of body weight at catastrophic failure was 665.7 ± 163.4% in the control group and 590.2 ± 117% in the treatment group (*p* = 0.055).

### 3.5. Modes of Failure

The modes of failure in each group were extremely consistent. In the control group, medial fracturing of the shaft was seen in nine of ten cases. For the nine cases, a fracture of the calcar occurred first, which then spread to the shaft (Figure 7).

In the treatment group, however, the cerclage reinforced the calcar and changed the lever arm, resulting in a lateral breakout of the distal tip of the prosthesis stem in nine of ten cases. Only in one case the cerclage detached prematurely and medial failure occurred as described in the control group (Figure 7).

## 4. Discussion

This biomechanical study investigated biomechanically the effect of cerclage banding distally to a clamshell fracture with an extent of 40% of the anchorage length of the stem. The described construct was tested versus a non-fractured femur model. The clamshell fracture was simulated using the same fracture model as in a previous study [30].

We were able to show that the application of a cerclage band located 3 mm distally to the fracture pattern is sufficient to regain almost full stability compared to a femoral model without fracture. No significant difference between the cerclage group and the native group could be demonstrated for any of the crucial studied parameters—loads and cycles at implant loosening and catastrophic failure. It can be concluded that in case of a clamshell fracture with such an extent, refixation of the medial wall fragment is not necessary, since the cerclage technique described above is sufficient to achieve adequate stability.

Our study follows up on several previous reports focusing on the medial femoral wall. In recent years, this topic has attracted increasing attention, mainly because of the increasing numbers of metaphyseal anchoring stems used for primary total hip arthroplasty [16]. Likewise, the good state of the medial wall has also been described as being extremely crucial for good clinical outcome after proximal femoral fractures [17,18,19,20]. However, the novelty of our study is the nature of the fracture pattern. In 2018, this particular pattern, called clamshell fracture, was added to the Vancouver classification by Huang et al. [22]. Until then, there was actually no distinction between simple avulsion fractures of the lesser trochanter and cases involving an integral part of the medial femoral wall [25]. Since the clamshell fracture belongs to the Vancouver B type fractures, according to the guidelines for their management the revision and implantation of a diaphyseal anchoring stem should be performed depending on the size of the medial wall fragment [22]. However, the question about how to proceed with the medial wall fragment has not yet been resolved. There is also an open question about how the comparatively simple placement of a cerclage band distally to the fracture pattern affects the axial stability. It has already been demonstrated that prophylactic cerclage wiring in the context of uncemented THA with well-fixed press fit stems leads to an increase in rotational stability and energy until failure [28]. Similarly, it has already been reported that in case of intraoperative Mallory type II periprosthetic fractures, cerclage wires lead to good mid- and long-term outcomes. No significant differences in Harris Hip Score, VAS Score, and X-rays Score could be described 6 months and 5 years postoperatively compared to patients without fracture after primary THA [37,38]. Therefore, the question arises as to whether cerclages placed 3 mm distally to the fracture lead to an increased resistance to failure in case of an intraoperative periprosthetic clamshell fracture. To address these questions, we performed the current biomechanical work implementing a progressively increasing cyclic axial loading protocol [31].

Our study has some strengths and limitations. As in all human cadaveric studies, only a limited number of specimens could be used. This limits the validity to actual clinical cases. Similarly, this study did not simulate the soft tissues including joint capsules, ligaments and muscles, which are largely responsible for the stability of the hip joint. Another point that has to be considered critically is the fact that the fractures were created with a saw and were therefore not fully comparable with real fractures. Unfortunately, it is not possible to recreate a real-life fracture in a biomechanical experiment, but as can be seen in Figure 8 that the fracture pattern we chose is comparable to the clamshell fractures described in other studies [21,24]. In addition, with our fracture design, a standardized fragment size could be determined for each stem size. Furthermore, there are no other existing biomechanical studies on clamshell fractures, so our results cannot be compared with those in the literature. The strengths of the study are evident in the pairwise testing of the specimens, resulting in excellent comparability between the groups. In addition, it was possible to use a new press fit stem for each specimen. Another strength of the study is that the created fracture model mimics the occurring fracture pattern when the intact specimens were tested (Figure 8). This means that the created fracture model is very close to reality, which further underlines the clinical relevance of the study. The modes of failure are also promising, as they were extremely consistent across each group. Both the machine data and motion tracking data could be evaluated and used in a combination. It is also very important to mention that a repeatedly used experimental setup was implemented in the current study, which secures generation of accurate and usable data [29].

In follow-up studies, the craniocaudal extent of the fracture model could be extended or the medial wall fragment could be additionally refixed to understand better the clamshell fracture behavior and to investigate which surgical treatment options might have the best biomechanical or clinical effects.

In summary, in case of an intraoperative proximal periprosthetic clamshell fracture, a diaphyseal anchoring stem should be inserted because—according to the literature—the stability of a metaphyseal anchoring stem is significantly affected [22]. A cerclage band placed 3 mm distally to the clamshell fracture is sufficient to achieve adequate axial stability, comparable with the native construct without fracture. It is also interesting to note that lateral failure of the femur occurred in nine out of ten specimens of the treatment group (Figure 7). Due to both the change in the lever arm and the strength of the cerclage band in this group, the point of maximum force application appears to be the lateral femoral wall. It should therefore be expected that with the application of a cerclage band 3 mm distally to a clamshell fracture resulting in 40% medial anchorage loss, almost full axial stability can be achieved, but the failure pattern differs from that of a femur without fracture. Regarding the medial wall fragment, it must be borne in mind that its refixation is also useful for other purposes, especially the ones related to the functional aspects of the results. On the other hand, it should be noted that even after successful refixation of the fragment, the traction of the iliopsoas muscle would keep on acting on the fragment of the lesser trochanter, which may lead to dislocation. Likewise, repeated gait cycles would result in stem loosening, and thus, the refixation could be only temporary [17]. Of course, if the fragment is easily accessible and the bone is not yet impacted, refixation should be considered to preserve the functional aspects of the medial wall fragment. If this is not the case, refixation can be omitted from a biomechanical point of view, since the axial stability of the stem is not impaired. Thus, a loss of time and the associated risks such as infection, intraoperative blood loss, or iatrogenic injury can be avoided.

## Figures and Tables

**Figure 1 bioengineering-10-01397-f001:**
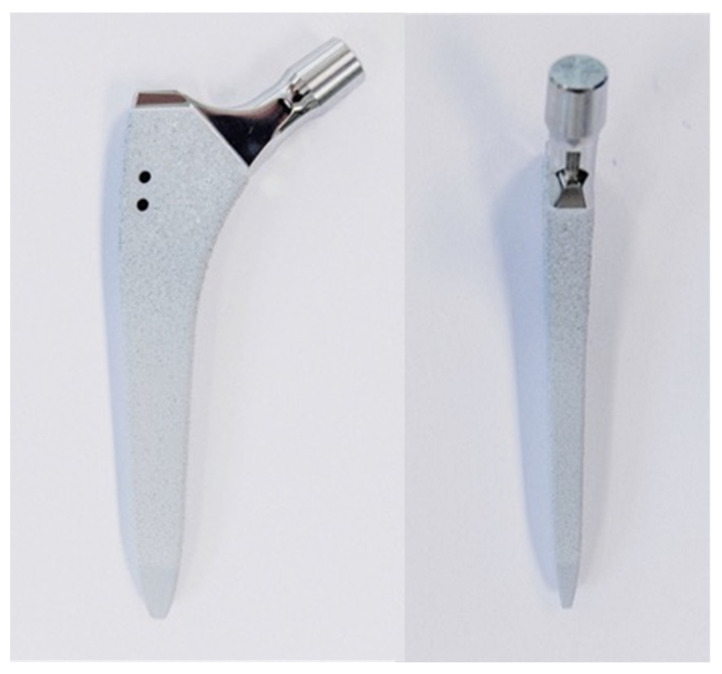
Photographs visualizing the used stem in anterior (**left**) and lateral (**right**) views: ANA.NOVA SL-complete, 127° CCD, double conus, calcium phosphate titanium plasma-coated press fit stem.

**Figure 2 bioengineering-10-01397-f002:**
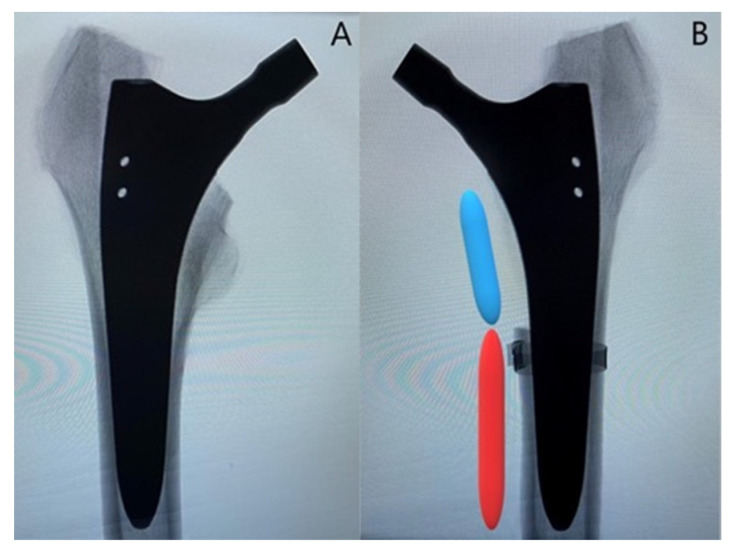
Representative anteroposterior X-rays of one instrumented specimen from each group: (**A**) specimen from the control group; (**B**) specimen from the treatment group with depiction of the created fracture model with an extent of 40% of the anchorage length of the stem (blue segment), while 60% of the anchorage length is preserved (red segment). Cerclage banding is visualized distally to the clamshell fracture.

**Figure 3 bioengineering-10-01397-f003:**
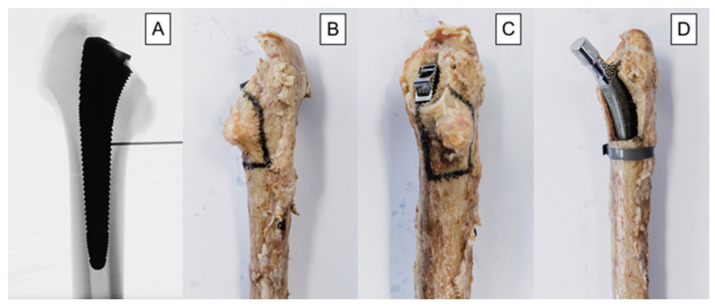
Depicted workflow for fracture model creation, based on individual stem size: (**A**) anteroposterior X-rays illustrating the definition of the craniocaudal extent of the fracture model. The desired 40% loss of medial anchorage was determined based on the serrations of the last rasp and measured using X-rays. The number of serrations, and thus the craniocaudal dimension of the fracture model, was calculated individually for each stem size. (**B**) dorsal-view photograph of an exemplified specimen with marked osteotomy lines; (**C**) dorsomedial-view photograph of an exemplified specimen with marked osteotomy lines; (**D**) anteromedial-view photograph of an exemplified specimen from the treatment group after instrumentation.

**Figure 4 bioengineering-10-01397-f004:**
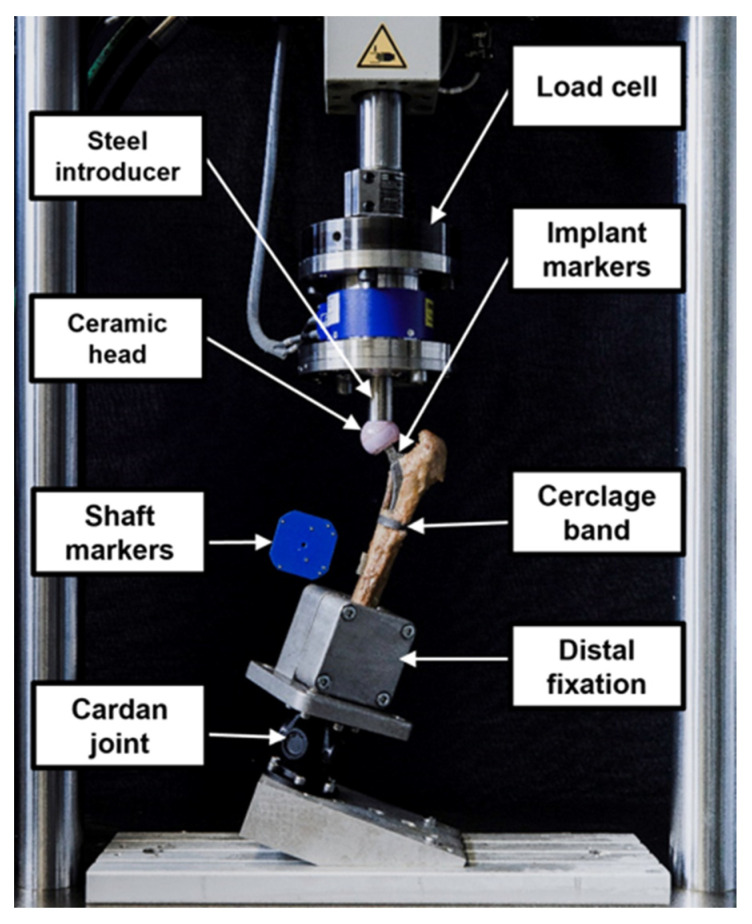
Setup with a specimen mounted for biomechanical testing.

**Figure 5 bioengineering-10-01397-f005:**
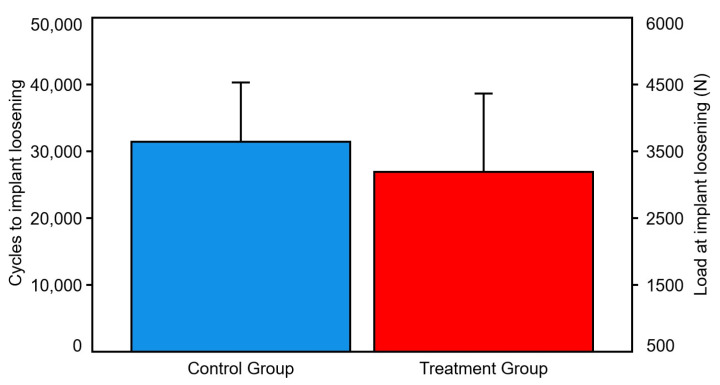
Cycles and load at implant loosening, presented for each group separately in terms of mean value and SD. No significant differences were detected between the groups.

**Figure 6 bioengineering-10-01397-f006:**
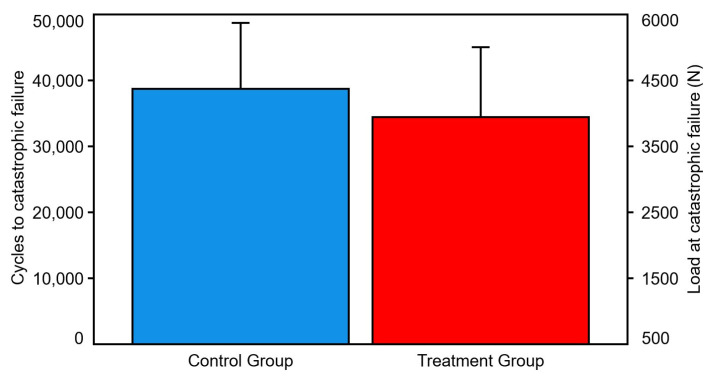
Cycles and load at catastrophic failure, presented for each group separately in terms of mean value and SD. No significant differences were detected between the groups.

**Figure 7 bioengineering-10-01397-f007:**
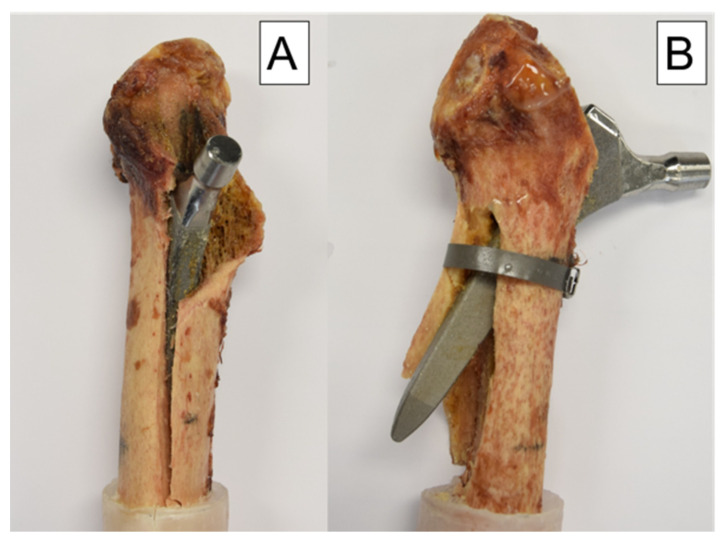
Modes of failure: (**A**) specimen from the control group with a medial fracture of the shaft and a typical clamshell fracture pattern proximally. (**B**) specimen from the treatment group with a lateral breakout of the stem because of the cerclage banding.

**Figure 8 bioengineering-10-01397-f008:**
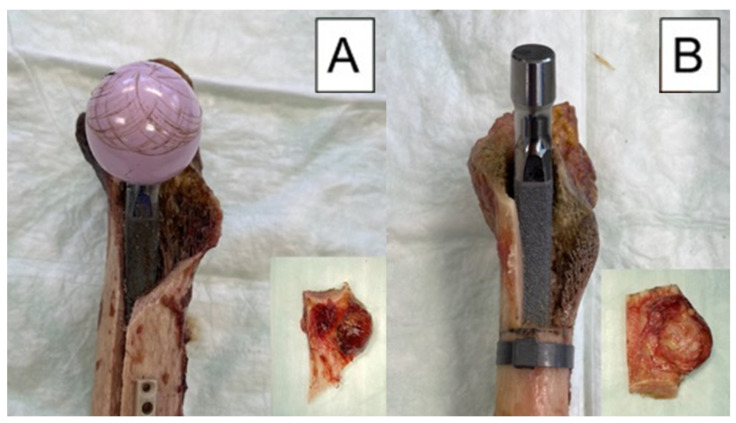
Exemplified modes of failure after biomechanical testing visualized with medial view photographs of specimens from each study group: (**A**) specimen from the control group after cyclic testing to failure, with its fragment as a result from the testing, shown in the embedded photograph; (**B**) specimen from the treatment group before cyclic testing, with its fragment as created with an oscillating saw, shown in the embedded photograph.

## Data Availability

The datasets used and/or analyzed during the current study are available from the corresponding author upon reasonable request.

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
