# Peer review of "The Effect of Cerclage Banding Distally to a Clamshell Fracture Pattern in Total Hip Arthroplasty—A Biomechanical Study"

_bioengineering, 2023, doi:10.3390/bioengineering10121397_

Round 1

Reviewer 1 Report

Comments and Suggestions for Authors

Authors present an interesting novel study.

The abstract is very clear on the purposes and objectives. Introduction section makes a correct description of the problem. Materials and methods are suitable, as well as the discussion of results. References are adequate and up to date.

Paper should be published.

Author Response

Thank you for taking the time to read my paper and thank you for the good review.

I hope I was able to give you an insight into the topic of clamshell fractures.

Reviewer 2 Report

Comments and Suggestions for Authors

The Authors aimed to investigate the effect of a cerclage band distally to a proximal periprosthetic femoral clamshell fracture versus a non-fractured femur after total hip arthroplasty. 

Abstract: experiment poorly presented. Conclusions are not clearly reported.

Methods: some data are nor reported.

Fig 1 does not add ay relevant information.

How were fracture done? According to the pictures, a linear fracture line might be extremely different from intra operative "real fractures". Please discuss

I would appreciate to see details about cerclage. Also, should it be placed before or after stem insertion?

Conclusions are not supported by results.

Many grammar and syntax errors. Please have the paper checked by an English native speaker.

Comments on the Quality of English Language

Many grammar and syntax errors. Please have the paper checked by an English native speaker.

Author Response

Thank you very much for reading my paper. I have taken note of your objections and hope that I have been able to answer them adequately.

  • the abstract is now adapted, and thus the project is hopefully summarized more precisely and simply
  • In fact, the results lacked representations of important variables, such as the multiple of the respective body weight. This was subsequently submitted
  • The creation of the fracture pattern was explained in the methods. The comparability to real-life fractures was added in the discussion. Thank you for this insightful comment.
  • The cerclage used is a standard cerclage as used worldwide. Basically, the idea is that this cerclage is placed after insertion of the diaphyseal anchoring stem to ensure better axial stability in the event of an intraoperative clamshell fracture. The text has been adapted slightly to make this clearer.
  • In principle, the conclusion from the paper is intended to show that axial stability is significantly improved with the help of the described cerclage in the case of an intraoperative clamshell fracture to the extent mentioned in the text. From a surgical point of view, this finding can save intraoperative time and the associated risks. Of course, a biomechanical experiment can never simply be transferred to the operating room. This was of course also pointed out in the discussion.
  • Also thanks again for pointing out syntax errors - I hope these could be fixed.

I hope my corrections could contribute to a publication.
I look forward to your reply.

Reviewer 3 Report

Comments and Suggestions for Authors

I was very glad to become acquainted with such interesting research. Authors made significant work, but, still, there are some remarks:

Line 92: 77±??? the value is missing.

So, authors point on quasi static loading, but there is no information about strain speed.

Lines 170-171: The schematic picture of cycle should improve the understanding the loading.

In the results Stiffness is mentioned, but in the introduction there is no description or equation for calculation of the parameter.

Statistics analyses presented for data, but it is not clear the amount of measurements. Yes, it was told (lines 202-204) that “10 samples per group were required”, but was it 10?

Did standard deviation of each group did not exceed 110% of the minimum difference in means between groups?

Figure 5 and 6 can be combined (e.g., 5a and 5b)

What about deviation of origin loading curves?

In methods stereometric measurement system was described, but there no such data in results. Or it was only considered for the assessment of stem loosening?

I advise to add conclusion to summarize main points.

Author Response

Thank you for taking the time to read my work, and thank you for your constructive criticism. As a surgeon, some of the objections were difficult to correct, but I hope that your concerns have been resolved.

  • Line 92: Error fixed
  • Strain speed was 15N/s

  • The reviewer noted that initial stiffness was mentioned but not included in the results. - As initial stiffness is of secondary importance from a surgical point of view, the results were not included in the paper.

  • Lines 202-204: Yes, 10 specimens were tested per group. I hope I was able to clarify this in the text. Thanks for the good point.

  • Limitations section:
    “..., only a limited number of specimens could be used. Indeed, with an underestimated sample size the study was underpowered, revealing no statistically significant differences between the groups.This limits the validity of to actual clinical cases….”

  • We anticipate that the reviewer is requesting information regarding differences between the applied double-peaked physiological loading profile and a standard sine curve. The applied loading profile reflects the course of the hip joint reaction force during walking, as demonstrated by Bergmann et. al (2001).

    Text change:
    “...featuring a dual-peak profile per cycle, reflecting the course of hip joint reaction forces during walking, as demonstrated by Bergmann et al. (2001) [33].

  • Exactly, we used motion tracking to determine the number of cycles up to a certain threshold where the stem moved relative to the bone. To do this, however, the relative movements had to be recorded continuously throughout the tests. In principle, stereometric measurement allows more precise measurements than, for example, machine data alone

We hope that we have been able to resolve the objections raised by the reviewer and would like to thank you once again for your constructive criticism.

Round 2

Reviewer 2 Report

Comments and Suggestions for Authors

Thank you to the Authors for their great efforts in the attempt to ameliorate their paper. It is now suitable for publication.

Reviewer 3 Report

Comments and Suggestions for Authors

The article revised as required.